# Demand-side load forecasting in smart grids using machine learning techniques

Muhammad Yasir Masood[1], Sana Aurangzeb[2], Muhammad Aleem[2], Ameen Chilwan[3] and Muhammad Awais[1]

[1] The University of Lahore, Lahore, Pakistan
[2] Computer Science, National University of Computer and Emerging Sciences, Islamabad, Islamabad, Pakistan
[3] Norwegian University of Science and Technology, Trondheim, Norway

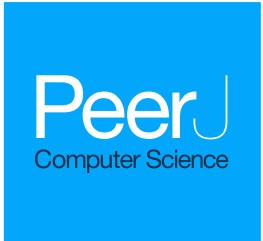

## ABSTRACT

Electrical load forecasting remains an ongoing challenge due to various factors, such as temperature and weather, which change day by day. In this age of Big Data, efficient handling of data and obtaining valuable information from raw data is crucial. Through the use of IoT devices and smart meters, we can capture data efficiently, whereas traditional methods may struggle with data management. The proposed solution consists of two levels for forecasting. The selected subsets of data are first fed into the "Daily Consumption Electrical Networks" (DCEN) network, which provides valid input to the "Intra Load Forecasting Networks" (ILFN) network. To address overfitting issues, we use classic or conventional neural networks. This research employs a three-tier architecture, which includes the cloud layer, fog layer, and edge servers. The classical state-of-the-art prediction schemes usually employ a two-tier architecture with classical models, which can result in low learning precision and overfitting issues. The proposed approach uses more weather features that were not previously utilized to predict the load. In this study, numerous experiments were conducted and found that support vector regression outperformed other methods. The results obtained were 5.055 for mean absolute percentage error (MAPE), 0.69 for root mean square error (RMSE), 0.37 for normalized mean square error (NRMSE), 0.0072 for mean squared logarithmic error (MSLE), and 0.86 for R2 score values. The experimental findings demonstrate the effectiveness of the proposed method.

## INTRODUCTION

Electric power is a crucial necessity and serves as the backbone of every region, directly influencing the economic condition (*Li, Ota & Dong, 2017b*). Recently, the smart grid has garnered significant attention as a viable solution to the global electric power deficit (*Son et al., 2018*). By efficiently managing energy, the smart grid has the potential to save money, leading to numerous projects aimed at addressing various smart grid challenges. It encompasses automation, communication, and information technology (IT) systems, allowing for the monitoring of power flows from generation to consumption points (*Dong, Qian & Huang, 2017b*). Consistent power generation that aligns with user demands

Corresponding author
Muhammad Aleem,
m.aleem@nu.edu.pk

on the demand side is imperative. However, load forecasting remains a persistent challenge due to the influence of various factors (*Li, Ota & Dong, 2017a*).

The grid controller must possess the capability to swiftly adapt to changes on both the demand and supply sides, ensuring efficient handling (*Amarasinghe, Marino & Manic, 2017*). An inherent issue lies in achieving equilibrium between the demand and supply sides (*Dong, Yassine & Armitage, 2020*). In light of the prevailing energy shortage, the contemporary era is increasingly gravitating towards smart grids. These systems address issues related to generation, distribution, and utilization by implementing diverse strategies across the power grid, utility, and demand side (*Ayub et al., 2019*). The old electric power grid lags in terms of control and dependability (*Ali, Adnan & Tariq, 2019*). IoT devices are pivotal in sensing real-time environmental data for smart electric grids, contributing to operations, maintenance, security, information retrieval, and safety management (*Rabie et al., 2020*). With the influx of substantial data from IoT devices, there is a need for geo-distribution and mobility support to minimize data latency. To prevent unnecessary data transmission directly to the cloud, edge servers are essential for improved processing, storage, and commuting of fog node computing (*Rabie et al., 2019*).

The extensive use of cloud data has led to various issues, including high data latency, low reliability, and network congestion (*Li et al., 2018*). The traditional grid operates as a one-way communication system, moving from the generation side to the consumer's side (*Hou et al., 2020*), however, the smart grid facilitates two-way communication (*Mujeeb & Javaid, 2019*). The data generated by electric power users is experiencing rapid growth, increasing in complexity, and ultimately transitioning into big data. Conventional data analysis models are no longer adequate to meet the demands of big data. This necessitates the development of a new data analysis model focused on assessing and processing large-scale data from the perspective of power users (*Wang & Sun, 2015*).

There are essentially four types of load forecasting in smart grids, as illustrated in Fig. 1. The first is short-term load forecasting, which allows one to predict load from 1 h to a week. The second type is medium-term load forecasting, enabling predictions for several weeks or months. The third type is long-term load forecasting, providing insights into loads for months or even years. The fourth type is very short-term load forecasting, allowing for predictions in the range of seconds to minutes.

Independent System Operators (ISOs) and other participants in the energy market employ short-term load forecasting (STLF) to determine pricing for the day-ahead market (DAM) and real-time balancing market (*Tavassoli-Hojati et al., 2020*). Load forecasting remains a critical yet unresolved challenge in smart grids, largely due to various influencing factors (*Li, Ota & Dong, 2017b*). Accurate load estimation is crucial for electric companies and consumers to prevent shortages (*Li, Ota & Dong, 2017b*). Decisions regarding load allocation for different areas during the daytime, nighttime, and special occasions hinge on factual load estimations (*Dong, Qian & Huang, 2017b*). STLF is imperative for power system management and scheduling, interchange evaluation, security assessment, reliability analysis, and spot price computation in daily power system operations.

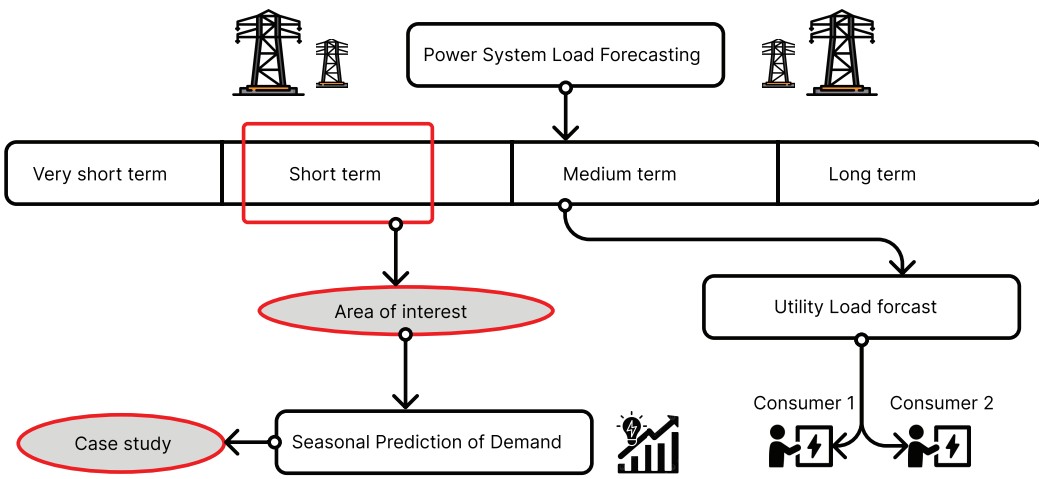

**Figure 1 Types of load forecasting in smart grids (*Raza & Khosravi, 2015*).**

While traditional machine learning-based short-term load forecasting exhibits proficiency in determining loads, particularly for understanding non-direct element fitting loads, artificial neural networks (ANN) and support vector regression (SVR) are the most widely recognized artificial intelligence techniques for developing prediction models (*Dong, Qian & Huang, 2017b*).

Price is not the sole parameter influencing load; various factors, including temperature, humidity, sunlight, *etc.*, can impact load prediction in an area (*Ayub et al., 2019*). Load diagrams are constructed using average power values, obtained by dividing instant power integration by the time interval, for each 15 min (*Chemetova, Santos & Ventim-Neves, 2017*). Load forecasting is crucial, especially for interconnected utilities that share their anticipated loads during peak hours, thus reducing the burden on individual utilities. Predicting load patterns allows utilities to understand consumer behavior in advance and aids in making financially viable decisions regarding future investments in transmission and distribution infrastructure. This foresight enables utilities to plan maintenance activities with minimal disruption to customers and fewer revenue losses (*Ali, Adnan & Tariq, 2019*).

While long short-term memory (LSTM) can effectively store one-dimensional sequence data for extended periods, predicting power load based on edge sensing data in a smart grid involves multi-dimensional time series, encompassing variables like temperature, weather, date, and more. In a smart grid context, future power demand depends not only on past power load data but also on various characteristics from edge devices, such as temperature, weather, and date time series. Machine learning provides efficient methods for predicting load data, including neural networks (NN) and SVR (*Dong, Qian & Huang, 2017a*). The choice of forecasting method depends on several factors, including the availability and reliability of historical load data, the geographic scope of the forecast, the accuracy of weather data, and the desired level of prediction precision. Consequently, selecting the appropriate load forecasting approach is contingent on the time frame of the prediction. It

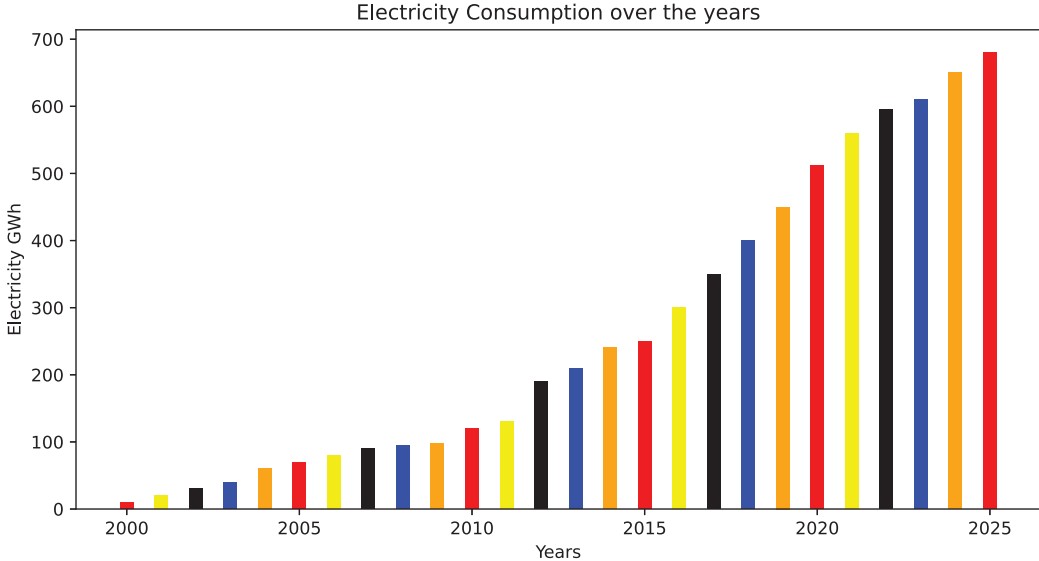

**Figure 2 Trends of electricity use around the world (*Tong et al., 2021*).**

is essential to consider the specific area, its temporal characteristics, user behaviors, and other relevant factors (*e.g.*, formal events, humidity, *etc.*).

Short-term load forecasting (STLF) faces four primary challenges, which can be addressed using various techniques. These issues encompass similar day prediction, variable selection, hierarchical forecasting, and weather stations. The similar-day technique treats load data as a collection of comparable daily load profiles. The variable selection method assumes that the load data behaves like a set of interrelated or independent variables. Conversely, the hierarchical approach regards the data as an aggregated load, influenced by changes in electric load at lower hierarchical levels. Finally, weather station selection involves identifying the most suitable weather data for integration into the load model. The trends in electricity usage around the world are shown in Fig. 2.

Many researchers have delved into STLF using neural networks (*Li, Ota & Dong, 2017b*; *Dong, Qian & Huang, 2017b*; *Dong, Yassine & Armitage, 2020*; *Khan et al., 2019*; *Hou et al., 2020*; *Fallah et al., 2019*). Previously, classical methods were prevalent worldwide for STLF (*Khan et al., 2019*; *Li, Ota & Dong, 2017b*). Achieving accurate load predictions leads to reduced operational costs, ultimately benefiting electric companies in terms of savings. This accuracy in STLF proves especially crucial in deregulated power markets.

When considering the variable selection method, load data can be condensed into a subset, but this approach doesn't apply to weather data. The alternative hierarchical method makes the false assumption of constant load fluctuations. The inclusion of weather data, as the fourth type, is imperative for accurate prediction. For instance, if we base our predictions on the assumption of sunny weather, there may be discrepancies if the actual conditions change. Conversely, if we predict the worst weather, it could lead to overloading issues in the smart home.

Load and price predictions are carried out using a variety of methods, and the challenge of training traditional forecasting techniques increases with larger input datasets (*Mujeeb et al., 2018*). To maintain a precise forecast of energy consumption, smart sensors/smart meters (SMs) are a must for a smart grid system (*Dewangan, Abdelaziz & Biswal, 2023*). Time series data is typically easier to forecast compared to local-level data due to its higher level of aggregate regularity. However, with the availability of smart meter data, local-level forecasting has become viable and proves useful for implementing efficient demand response strategies.

Medium-term load forecasting is becoming increasingly crucial in demand-side management planning, storage maintenance, and scheduling (*Han et al., 2018*). External variables like weather, days of the week, energy costs, and others significantly influence electric load, and many of these are challenging to forecast especially long-term weather conditions. Despite extensive research, reliable electric load forecasting in smart grids remains a challenging problem (*Zheng et al., 2017*). Electric load forecasting is often a univariate time series forecasting problem, which tends to be more challenging than its multivariate counterpart.

In the current state of the art, there are some notable limitations in electricity data modeling (*Dong, Qian & Huang, 2017b*; *Amarasinghe, Marino & Manic, 2017*; *Khan et al., 2019*; *Li, Ota & Dong, 2017a*, *2017b*). These include issues of repetitions and redundancy in the data models, which a few articles have addressed by identifying solutions to handle abnormal data. Another limitation arises when a user divides or classifies the information. Determining the required level of clustering in advance, and more critically, considering the distribution density of electricity consumption among users, is often overlooked. Additionally, modeling the information characteristics may not be fully extracted, or the data may not be thoroughly refined, and the relationships between these characteristics may not be entirely considered. Finally, many researchers tend to rely on a single classical model in which data features are not fully leveraged, leading to potential overfitting issues (*Li, Ota & Dong, 2017b*, *2017a*).

In this research, the following techniques are used to make this research unique and impactful for others.

- Fine-tuned model is used to prevent overfitting. An extensive feature engineering is performed for the selection of the most relevant data and map the data into a correlation graph.
- Incorporating additional weather features (temperature, humidity, UV index, dew point, wind speed, cloud cover, visibility).

## RELATED WORK

In *Li, Ota & Dong (2017b)*, the authors introduced an IoT-based load forecasting scheme with a two-step approach specifically tailored for daily load prediction. They employed deep learning techniques to extract features from various influencing factors. Additionally, the authors proposed an analytical method to establish the relationship between these factors and electrical load. However, one limitation of this article is the substantial amount

of data that must be transmitted from the system through IoT devices and smart meters. One possible solution to address this challenge is the implementation of a fog layer, which operates with low latency between the cloud and the IoT devices. It is worth noting that the authors did not incorporate weather data into their prediction model, which could have a significant impact on load forecasting accuracy.

In *Dong, Qian & Huang (2017b)*, the joint convolutional neural network (CNN) and K-means clustering algorithm scheme are proposed for the prediction of load based on hourly data. The CNN model is used for feature extraction and selection. The K-means algorithm is used to segment the data into different subsets as they have been implemented in the large natural environment for faster and more efficient construction models. The authors deal with a massive amount of data, many parameters, such as cluster numbers, and learning time for the CNN model. For that purpose, the authors had to train many subsets to construct the model, which consumes a lot of time and computation resources.

In *Li, Ota & Dong (2017a)*, a deep learning-based short-term load forecasting mechanism is proposed. The authors transformed the load forecasting module into an image dispensation and designed the two-division deep neural network to extract features from input data. For load forecasting, the authors proposed a multi-layer neural network for the prediction of the load. Their proposed deep learning-based short-term forecasting (DLSF) method also influences temperature, humidity, and wind speed compared with other forecasting schemes. The limitation of the approach was that they only differentiated their result with the SVM method, which already has the over-fitting problem.

In *Dong, Yassine & Armitage (2020)*, the authors presented a forecasting system for daily electricity consumption based on the daily load curve structure. The prediction through this image-based system is better for regular days than the peak days load prediction. The authors implemented three algorithms for prediction: i) Random Forest, ii) XGBoost, and iii) Cubist. The drawback of their approach was that it is only feasible for regular days in the week, furthermore, the authors did not consider load influence factors in their system, which can impact load forecasting.

In *Ayub et al. (2019)*, the HFSEC SVM classifier method proposes accurate load forecasting. Their model is based on two stages, which are feature engineering and classifier adjustment. For the feature extraction and selection, they used XGBoost and decision tree classifier (DTC) from the input data. After selecting data, they found that the features had redundancy, so they removed the redundancy through a recursive feature eliminator (RFE). The drawback of the study is its high computational complexity and slow processing of data and also it processes the data in a week-by-week sequence. It was tough to find the exact parameter values of the cost penalty, kernel parameters, and motivation loss function. They had considered the real load value for evaluating their system and claimed to get 98% accuracy.

In *Ali, Adnan & Tariq (2019)*, the authors proposed two approaches: the first was how a system detects the overloading problem and how to tackle renewable energy resources in smart grids. The second one was load prediction through a fuzzy-based system. Their approach provided a better balance between the supply side and the demand side. For load prediction, authors have considered the humidity and temperature which are the most

influential factors. The authors claimed that the controller-based work was the first work in that load forecasting area. Hence, we can better manage the resources and find a better balance between the supply and demand sides. They tested the work through nine buses, and it showed effectiveness. It took a significant number of repetitions to get the best answer for minimizing the overall functioning cost of renewable energy. It results in a variance between the total load and the predicted load.

In *Rabie et al. (2020)*, an outlier rejection methodology describes how the authors tried to remove the outliers effectively on the big data. They proposed the two-tier architecture load forecasting scheme, hybrid outlier rejection methodology (HORM), which contains two phases. The first one is fast outlier rejection (FOR), and the second one is accurate outlier rejection (AOR). Before implementing the HORM method, the authors used the fuzzy-based feature selection method to remove irrelevant features. The authors got better results after effectively removing the outlier. The HORM introduced the improved effectiveness of the load prediction method. Still, the drawback of their approach was that its run time was extensive, and without weather forecasting, it created issues in both FOR and AOR outlier rejection. The map-reduce method's real-time decision-making was another disadvantage. A real-time decision-making algorithm must be used.

In *Kumar & Yan (2023)*, authors introduce a comprehensive predictive demand-side management (PDSM) approach with two main components. The first component involves predicting the day-ahead shiftable load by integrating the stacked long short-term memory (SLSTM), artificial neural network (ANN), and shiftable equipment matrix (SEM) modules. The SLSTM module forecasts day-ahead load variations (%) using load time series data segmented by the percentile-based method. The ANN module, with dynamic feature selection, predicts day-ahead load (kW) using K-means based on historical meteorological and load data. The SEM module determines the average percentage of shiftable equipment load using electric data from neighboring NGs. The second component focuses on user-centric multi-objective optimization through load shifting. A user-centric Mixed Integer Quadratic Programming optimization model is developed to shift the predicted shiftable load, minimizing energy costs and discomfort for the user. Their results indicate that the SLSTM predicts variations with an R2 of 97.6%, MAPE of 9.7%, and MSE of 0.0274%, while the integrated approach predicts shiftable load with an R2 of 95.84%. Additionally, daily energy costs can be reduced by up to 5.17% through user-centric multi-objective optimization. However, the proposed approach cannot be used in the big smart grids due to its computational complexity also, comparable load forecasting results to other schemes were missing.

In *Hou et al. (2020)*, the authors proposed a privacy-preserving implementation on Edge-Fog-Cloud for short-term load forecasting. The authors had moved the machine learning workload to the distributed smart meters and then transferred this data to the central cloud, which takes on the heavy burden of load transmission and keeps the local data stored on the smart meters. They also proposed protection schemes for the data stored on the smart meters from attackers. The cloud obtained the regional load forecasting of the whole fog area network. They had implemented this on real-world smart meter data sets and got an accurate prediction. Their research drawback was that they focused on the

smart meters to prevent attacks and interference from the cloud and did not get comparable load forecasting results to other schemes.

In *Mujeeb & Javaid (2019)*, the authors proposed two deep learning (DL) based models: ESAENARX and DE-RELM. They used these methods to get only the side influences of demand and prices on each other and thus can capture interdependencies in the market. The feature selection in ESAE improved the extraction of features that can help in effective load forecasting. The proposed method had a lower MAPE and RMSE than the traditional methods. This approach was feasible for the micro-grids and, thus, cannot be used in the big smart grids due to its computational complexity.

In *Xie & Hong (2017)*, the authors proposed a valid variable selection method for the probabilistic load predictions. They also presented a holistic method (HoM), which selects the relevant variable load prediction. The authors also contributed an analysis method for comparing the holistic method with the heuristic method (HeM), in which variables were selected by neglecting forecast errors. When variables in the model require realism and the same perspectives are considered suitable. HoM slightly outstrips but does not lead Leader HeM in the year of probabilistic predictions of the future load.

In *Zheng et al. (2017)*, LSTM based RNN recommend solving the problem of STLF. Long-Short-Term-Memory has fulfilled the forecasting of complex difficulties. The models are regulated by an internationally recognized short-haul passenger data record and a long-running electronic data set. Experimental results show that LSTM-based load methods can be far more advanced than old-fashioned ML forecasting methods to tackle short-term electrical load problems.

In *Liu et al. (2020)*, the data-imaging conversion (DIC) proposed better predicting the smart grids' load. The DIC scheme extracts meaningful features from the edge sensing data. Empirical mode decomposition (EMD) uses load influence factors like temperature, weather, and date data. The proposed scheme improves the training speed by 61.7%, reduces RMSE by at least 32.9%, and improves the prediction correctness by 1.4%, which can ensure the standard construction and life of society, effectively reduce the cost of power generation, and improve economic and social benefits.

In *Jeyaraj & Nadar (2021)*, authors presented demand-side energy management focusing on designing and developing computer-assisted residential energy management through forecasting using a deep learning algorithm. Model optimization is achieved through a pooling-based deep neural network (PDNN). The PDNN model is implemented in the TensorFlow platform. The proposed deep learning model demonstrates superior performance, surpassing support vector machine by 9.5% and 12.7%, deep belief network by 6.5% and 9.5%, and neural network auto-aggressive integral moving average by 20.5% and 8.5% in terms of energy forecasting accuracy and mean absolute error, respectively. However, more external features, for instance, weather information, temperature, *etc* are missing. The optimal network structure for completely avoiding the overfitting point is necessary for further analyzing the network size.

In *Khan et al. (2019)*, DL techniques propose to forecast load prediction. The proposed method was based on data extraction and then selecting the data and addressing the data classification. The feature selection algorithm was RF and mutual information, with kernel

principal component analysis (KPCA) used to extract features. They used CNN for the classification of data. Later, the data was normalized for training and testing sets. The proposed scheme was compared with the benchmark schemes and claimed to have better accuracy. The limitation of their approach was finding the optimal number of parameters for which they have used to test and train their model.

*Hernández et al. (2014)* developed an ELF-based ANN methodology for smart grids, which consits of three primary stages: segmentation utilizing K-means classification, a self-organizing map approach for pattern recognition, and demand forecasting within individual clusters (*Alquthami et al., 2022*). They validated the ANN model using real-time data from a Spanish corporation, employing periodic values for model training. The identification of irregular energy usage patterns in buildings is achieved through the application of outlier detection and clustering analysis techniques. This framework demonstrated superior performance compared to benchmarks utilizing generalized regression NN and radial basis function NN. The sustainability of the SG relies on its ability to continuously generate electricity based on usage. However, important parameters such as humidity, UV index, dew point *etc.* are missing that can be further investigated.

In *Ahmad & Chen (2018)* author utilized three diverse ML frameworks for medium-term load forecasting and long-term load forecasting in the smart grid. They employed a nonlinear ANN consisting of ada boost, multivariate linear regression, and auto-regressive exogenous multivariate inputs framework. The researchers categorized the load into three intervals based on aggregated exhaustive consumption metrics: 1 month ahead, seasonal perspective, and 1 year ahead. These models improved predictability while accurately defining energy differences, modifications, and forthcoming energy prediction prospects. Due to its superior predictive capability, the Ada boost model outperformed the other models.

In this (*Alquthami et al., 2022*) authors utilizes several ML algorithms such as logistic regression, SVM, NB, decision tree, KNN, and neural networks to analyze the performance. The aim is to present a comparative analysis of ML algorithms for short-term load forecasting regarding accuracy and forecast error. The study concluded that among other algorithms, the decision tree classifier comes up with better results. For this, they utilized enhanced decision tree after integrating fitting function, loss function, and gradient boosting for fine-tuning the variables. The results indicates that the proposed algorithm provides better forecast results.

In *Arumugham et al. (2023)* authors presented the demand-side management strategy for the smart grid treating demand response from wind power and solar power generation. The overall operational cost of the microgrid was evaluated across various scenarios. Also, deep learning-based prediction models were developed to predict wind power and solar cell power generation. Consumers are anticipated to participate in demand response programs based on incentive payments aimed at consumption management. To address the model the MOACO method was employed. Simulation results indicated that the integration of demand response, coupled with mitigating production losses caused by wind power and solar power uncertainties, could lead to reduced operating costs. However, optimal energy conservation and management should be considered to ensure efficient operating conditions within the smart grid.

**Table 1 Summary of the related work.**

| Reference, Year | Methodology | Strengths | Weakness |
|---|---|---|---|
| *Li, Ota & Dong (2017b)* | • IoT-based load forecasting scheme | • Two-step prediction of load scheme<br>• Increases the forecasting for total daily consumption | • Over-fitted network<br>• Congestion of data on DCEN network |
| *Dong, Qian & Huang (2017b)* | • Joint CNN and K-means algorithm | • CNN enhance the performance of tasks with less feature | • Large subsets consume Lot of time<br>• Non-inclusion of weather data |
| *Li, Ota & Dong (2017a)* | • Deep learning-method for prediction of short-term load<br>• Transformed the load forecasting task into an image problem | • Perform accurate clustering on data using CNN<br>• DLSF method performs well in both accuracy and efficiency. | • Overfitted model<br>• Complicated model for prediction |
| *Dong, Yassine & Armitage (2020)* | • They implemented three algorithms for prediction 1. Random forest 2. XGBoost 3. Cubist | • The proposed scheme can predict odd days load effectively | • Feasible for the regular days<br>• Non-inclusion of impact factors |
| *Xie & Hong (2017)* | • Holistic method (HoM) to select a subset of relevant variables<br>• Analysis framework between (HoM) and (HeM) | • Proposed (HoM)slightly outperforms but does not dominate (HeM) skill of probabilistic load | • Not applicable on other states<br>• Non-inclusion of weather data |
| *Zheng et al. (2017)* | • Long-short term memory (LSTM) based recurrent neural network (RNN) | • Better finding temporal sequences<br>• Forecasting accurately the complex nonlinear load | • Complicated model for load prediction<br>• Non-inclusion of impact factors<br>• Tested on a small data set |
| *Liu et al. (2020)* | • A data imaging conversion scheme proposed to extract the features | • Used DIC and EMD features<br>• Got accurate load forecasting | • Complex model for prediction<br>• Not got comparable prediction than other models |
| *Khan et al. (2019)* | • Deep learning (DL) technique introduced<br>• Grey correlation-based random forest (RF)<br>• Mutual information (MI) performed for feature selection | • Got better subsets of data in the extraction and selection<br>• Got comparable load forecasting result | • Non-inclusion of impact factors<br>• Over-fitted-model |
| *Amarasinghe, Marino & Manic (2017)* | • CNN for individual building level load forecasting<br>• Targeted very short-term load forecasting<br>• A multiple-layer neural language network used for the final regression task | • CNN used for features engineering at the individual building level | • Non-inclusion of weather and factors<br>• Compared results with traditional methodsOver-fitted model |

In this study (*Habbak et al., 2023*) authors have conducted a comprehensive review of cutting-edge forecasting methods, encompassing traditional, clustering-based, AI-based, and time series-based techniques, and offers an assessment of their effectiveness and

**Table 2 Summary of the related work.**

| Reference, Year | Methodology | Strengths | Weakness |
|---|---|---|---|
| *Rabie et al. (2020)* | • Proposed outlier rejection<br>• Big data outlier rejection<br>• Proposed the three-tier architecture in place of two-tier architecture<br>• HORM consists of FOR and AOR | • Filtering data from noise provides better predictions<br>• Outlier rejection in the forecast can make a better forecast | • More significant time required for the prediction<br>• FOR and AOR methods have drawbacks<br>• Lack of real-time decision-making in map-reduce |
| *Mujeeb & Javaid (2019)* | • Proposed ESAENARX and DE-RELM9 (deep-learning based models)<br>• The feature selection ESAE improved the extraction of features | • ESAE significantly improves the quality of extracting feature<br>• Proposed models efficiently capture price-demand trends in energy big data | • Feasibility of proposed methodology only for microgrid<br>• Not feasible for the natural environment<br>• Non-inclusion of impact factors |
| *Ayub et al. (2019)* | • XGBoost and DTC from the input data for feature extraction<br>• Removed redundancy through recursive feature eliminator | • Through well-selected features through a hybrid approach, gain better accuracy up to 98%.<br>• Outperforms from traditional schemes | • Hard to find the exact parameters of cost<br>• The complexity model for prediction<br>• Weak processing of uncertain data |
| *Ali, Adnan & Tariq (2019)* | • Proposed renewable energy resources in the smart grid<br>• Fuzzy-based system for load forecasting | • The controller senses any disturbance and requests to controller<br>• Inclusion of weather data | • Complex model for prediction of load<br>• Did not include variation in load |
| *Rabie et al. (2019)* | • Proposed the IoT based enhanced smart electrical grid<br>• FBFS based feature selection method | • NB classifier as a load prediction method correctly trains on features subset resulted from FBFS<br>• FBFS improves the forecasting | • Issues in the feature extraction and selection<br>• Non-inclusion of weather data |
| *Li et al. (2018)* | • Proposed the XGB-ARMA model for short-term forecasting<br>• Implemented a K-means algorithm first with the help of Preto principal | • Local computing<br>• Real-time decision making of the proposed scheme | • Non-inclusion of weather data<br>• Need a lot of time for combining these methods |
| *Hou et al. (2020)* | • Focused on short-term prediction<br>• Protection of data from the attacker in smart meters | • Focused on the attack prevention of smart meters<br>• Got comparable results for short-term load forecasting | • Non-inclusion of weather data for prediction<br>• Predicted results are not so much accurate |

outcomes. The objective of this research is to ascertain the LF technique most apt for particular applications within smart grids (SGs). Results suggest that AI-based LF techniques, leveraging ML and NN models, exhibit superior forecasting performance compared to alternative approaches, yielding lower overall root mean squared (RMS) and MAPE values. Their findings indicate that AI-based LF techniques, incorporating ML and

NN models, have demonstrated the most favorable forecast performance among the methods investigated. These techniques have also exhibited higher overall accuracy, as measured by root mean squared error and MAPE, compared to other applied LF techniques. Furthermore, the combination of ML models with statistical approaches might enhance the precision and effectiveness of forecasting methods. Leveraging real-time data and deploying advanced sensor technology could further improve the capability to accurately predict and respond to changes in load demand. Additionally, incorporating distributed energy resources and considering the integration of renewable energy sources into forecasting models can offer a more comprehensive and sustainable approach to LF in SGs systems.

It is clear from the prior studies that DL approaches are valuable. Thus, the current effort focuses on the application of CNN techniques to proposed model consists of two level of forecasting. Moreover, fine-tuned model is used to prevent overfitting. An extensive feature engineering is performed for the selection of the most relevant data and map the data into a correlation graph. Also, incorporating additional weather features (temperature, humidity, UV index, dew point, wind speed, cloud cover, and visibility). The summary of the above mentioned literature work is summarized in the Tables 1 and 2, covering the strengths, weakness and load forecasting scheme used.

## THE 3-TIER LOAD FORECASTING ARCHITECTURE

This section discusses the proposed system architectural plan, which is depicted in Fig. 3. The proposed method consists of a three-tier architecture for smart grids. The electricity obtained is from resources for its production and distributed the load per the requirements of resources for its consumption. There are several different forms of electric city generation resources, including solar power, wind power, nuclear power, and hydropower. The electricity distribution lines connect these generation resources to the generation companies. The smart electrical grid is the automation and network communication system that balances the demand and electricity supply. The motivation behind using smart grids is to balance the demands, needs, and supply automatically. The smart electrical grid also has the advantage of requiring less human interaction. The load goes to the distribution companies and distributes load according to the smart homes and schools' needs and factories. Here are some other factors: In addition to becoming prosumers who can contribute to the system by producing electricity, our home customers can also be prosumers. That is why we use two-way arrows there, and it means that we can also provide a load to these utilities and get the electric load from them, and that load can also be used in the system.

Smart homes, schools, and factories are connected through communication lines with smart meters through wireless fidelity. Smart meters can be defined as an electrical meter that measures all the load flowing in and out of our home at brief time intervals. The smart meters can stay inactive for 24 h and capture the electricity in and out of our schools, universities, and factories. The key benefit of using smart meters is that we do not need to capture the electric meter reading manually every month. Furthermore, smart meters can capture the data in a real-time environment. With the usage of smart meters, we can utilize

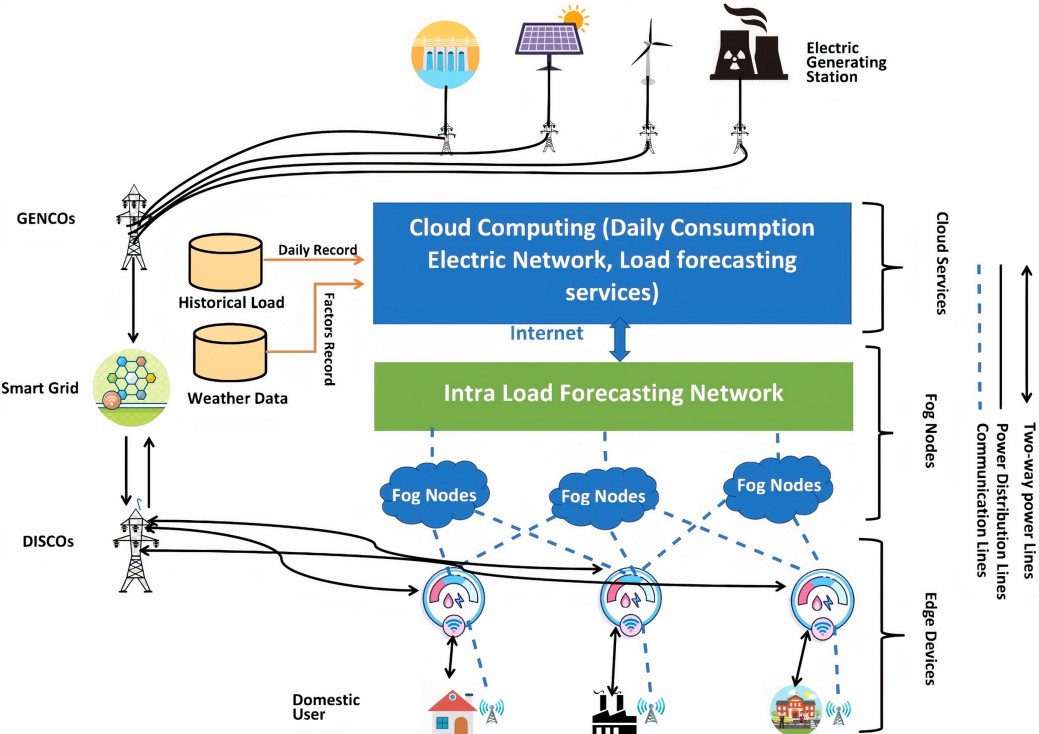

**Figure 3** 3-tier load forecasting scheme in the smart grids.               

efficient use of electrical resources. Now we talk about the disadvantages of using smart grids. The first one is the deployment cost, which can make it more costly than the other methods. Furthermore, the other one is that we have to ensure the privacy of smart meters.

The smart meters stay active for 24 h and capture a lot of data, called big data. The critical problem of big data is its handling because of redundant and outlier features in the data. Many schemes use the big-data for load prediction. However, there are a few limitations such as it can be seen from the literature that the researchers did not utilize weather data, which can have a significant impact on load forecasting.

Taking into consideration data, there exists another issue, which is an over-fitted model. For better performance and to avoid over-fitting, a system must make subsets of the data and make the data sorted. The over-fitted model usually gives better performance in the training data set and shows degraded performance in the natural environment. That is why there is a significant performance issue in the over-fitted model.

In this study, to avoid the over-fitting problem, firstly, we collect huge data and later extract and select the past day's best possible features. The key benefit of this is that there would be no conjunction of the data. Furthermore, we can get the sorting features so that we can get a better result. This extensive big data can be handled on the cloud, where it can be used as comprehensive data. Later, a Daily Electric Consumption Network (DCEN) model on the cloud, where can predict the daily load.

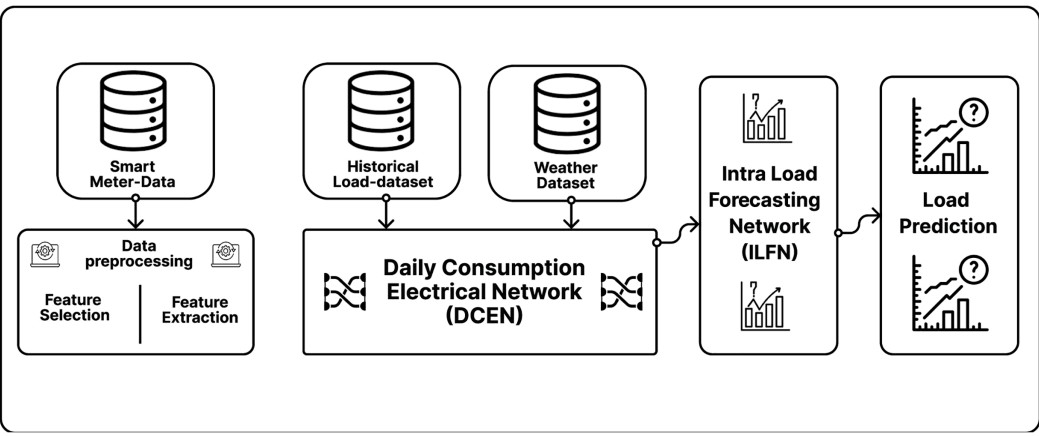

**Figure 4 Flow of data in three-tier load forecasting scheme.**

Two other data sets are used in the proposed approach: historical load and weather data—the city's historical pack on the hour's granularity. Furthermore, the weather data used here is for the future load prediction of that specific area. However, one network's basics (DCEN) cannot be predicted because of its small in number that could be insufficient for the users and the load companies. The significant estimation can lead to a system of overloading problems. Here, other factors (*e.g.*, user's behaviors and the political and social impacts on the user and issues like prices) can be added. From the DCEN network, valid input can be achieved for the dual system, which is the Intra-Load Forecasting Network (ILFN). ILFN network can check the daily load variations. This study aims to use the deep neural language network on the ILFN network to extensively handle the data based on all the inputs on this network to get better load predictions for the future.

In *Li, Ota & Dong (2017b)*, the authors used the two-tier cloud architecture of the cloud. They were getting the data through the IoT devices and collecting the data using smart meters. The proposed two-step load forecasting scheme was DCEN (*Li, Ota & Dong, 2017b*) and ILFN (*Li, Ota & Dong, 2017b*). DCEN is not helpful for load companies and consumers, but it provides significant input to the ILFN network. The DCEN cannot predict the load accurately individually. This proposed study uses two data sets for prediction in this scheme: the load data set and the factor data set (like temperature and humidity). The pre-processing of the data is necessary before entering the system, and getting helpful information is also required for this system.

## Detail working of the proposed scheme

In the data flow (Fig. 4), firstly, the smart meters electric city data from the London data set *Daignan (2014)* is taken, where the total records are 1,045,876. The actual problem is the cleansing and pre-processing of the extensive data set by removing the null values. There is a total of seven main features in the daily electric city data set of which three main features from this data set were selected: date, total energy-sum, and the Lclid (which defines the use of electric data). Later, the historical load record data (*Daignan, 2014*), were pre-processed. There are many irrelevant factors in this data set. Further, the dataset for

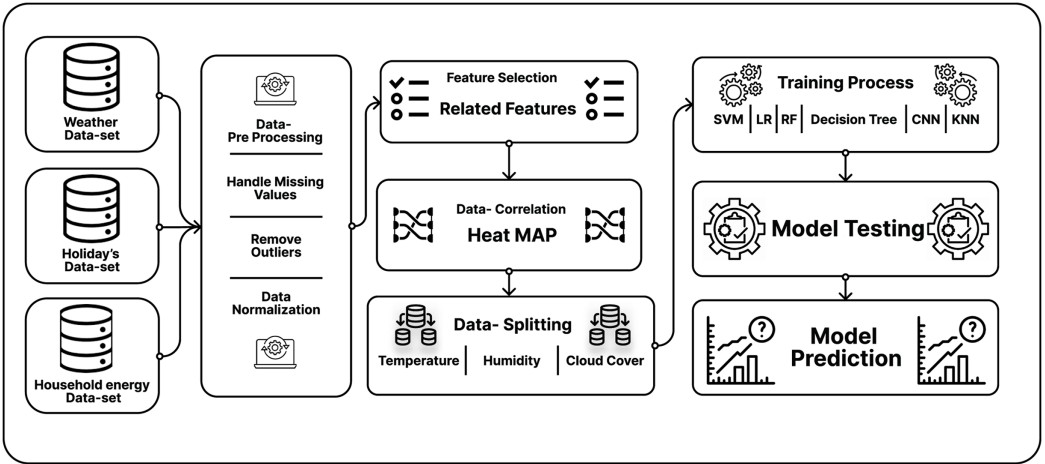

**Figure 5** Testing & Training diagram for DCEN and ILFN.

training and testing was split. For this, the study used 80% data for training and 20% for testing. The graph shows that the model performs well in training and tries to classify the data in the testing graphs. It indicates that the model didn't show the overfitted behavior on this data.

In the third section, the weather data set (*Daignan, 2014*) is taken into consideration, which plays an essential role in predicting the load in a specific region. The weather data to the energy data set is mapped and removed the irrelevant factors from the weather data set. There is a total of 883 weather data set samples in the weather data set. Most of the weather features primarily used in the literature article (*Dong, Qian & Huang, 2017b*; *Li, Ota & Dong, 2017b*; *Hou et al., 2020*) were temperature and humidity. In contrast, in this research, many other weather features were used, directly impacting the load prediction for the next day. This research contains features such as Visibility, humidity, wind speed, perception time, sun-rise time, cloud cover, and temperature.

In Fig. 4 we have presented the data flow in our proposed scheme where two models of forecasting scheme were implemented. In the first model, which we name daily consumption electrical network (*Li, Ota & Dong, 2017b*) considering the holidays and the weather data for the load prediction, also presented every entity that is included in that model.

In Fig. 5, holiday data and the weather data were taken into consideration for the pre-processing of the data; where the missing values from the data were removed from the outliers of the data and then normalized data. Then, correlation analysis of these features with the energy data was conducted. In the third step, the best features were selected which do not show co-relation with each other. The best-fitted features for our model implementation were selected. In the literature (*Dong, Qian & Huang, 2017b*; *Li, Ota & Dong, 2017b*) showing over-fitted behavior just because the authors were not selecting the best-fitted features. So, as a result, their proposed model shows over-fitted behavior.

In Fig. 5, next the household energy (*Daignan, 2014*) for our load prediction model in the intra-load forecasting network (*Li, Ota & Dong, 2017b*) for the prepossessing of the

**Table 3 Data set samples and descriptions (*Daignan, 2014*; *TRUDIE, 2014*).**

| Name | Total numbers of samples | Data set type |
|---|---|---|
| Daily load data (*Daignan, 2014*) | 1,045,876 | Smart meters data |
| Weather-data (*TRUDIE, 2014*) | 883 | Temperature, humidity, clouds |
| Hourly historical data (*Daignan, 2014*) | 21,166 | Historical load data |
| Public holidays (*Daignan, 2014*) | 26 | Holidays in 1 year |

smart meters data is taken. In which total of nine features in this data set (*Daignan, 2014*), where selected against the best features.

## Data set details

Most of the work on this problem has used a private data set. So, finding a valid data set for experiments is also a challenge in this problem. This study uses the smart meters data and the historical data of London from 2011 to 2014 (*Daignan, 2014*). The weather data and the holiday data were taken from another resource (*Daignan, 2014*). This data set covers the electric and weather data set of London.

## Data set samples

Three types of data are used: The first is on the weather data (*Daignan, 2014*), which plays a vital role in short-term load forecasting. The weather data (*Daignan, 2014*) has 883 samples. The second one is the historical load data, being used to predict the load. The historical data (*Daignan, 2014*) has 21,166 examples with a granularity of 1 h. Thirdly, the smart-meters data (*Daignan, 2014*), which is used for short-term prediction, also we need the holiday records of the specific region. In our case, the holiday data (*Daignan, 2014*) has 26 records. Finding of accurate data set is a big challenge as most of the work on electricity is not available online. The data set samples and description are shown in Table 3.

## Exploratory data analysis of data

Most of the work on this problem is on a private data set. So, we have got the load record of London from open-source (*Daignan, 2014*) and got the weather and the holidays record from the dark sky (*Daignan, 2014*). Then, we just mapped the weather data on the energy data set and made the exploratory data analysis (EDA) on the data set.

This section covers the data-set analysis and the trends. Figure 6, depicts the direction of the data, which tells the maximum temperature, minimum temperature, and average energy, respectively, with the orange, pink, and blue lines.

In Fig. 6, shows that when the maximum temperature goes high, and the minimum temperature goes low the average energy demand increases. Because of using the data set of London, there are different kinds of electric user trends. The unit of temperature used in this study was Celsius, and the energy unit was Kwh. This figure shows the movements of data from 2011 to 2014.

There is the exploratory data analysis of humidity *vs* average energy. Humidity is an essential factor in the prediction of the load. The trends of humidity and energy are shown

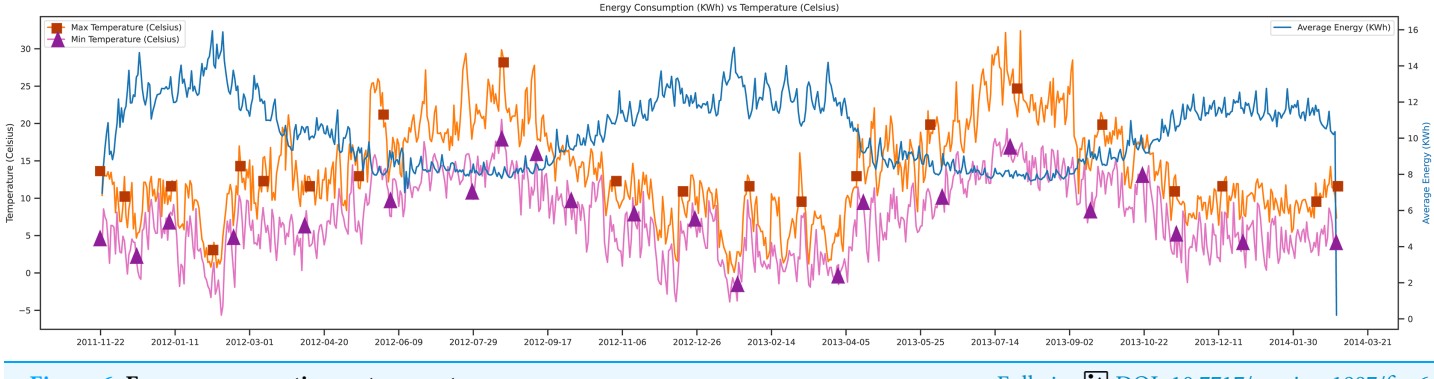

**Figure 6 Energy consumption *vs* temperature.**

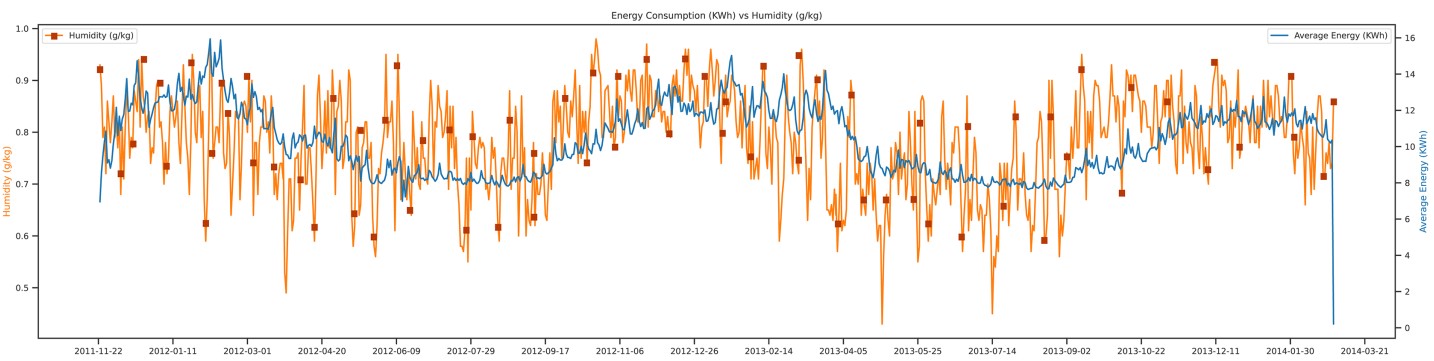

**Figure 7 Energy consumption *vs* humidity.**

in Fig. 7. Humidity is the water vapors in the air, and it depends on the current temperature of the day. Figure 7 analyzes the energy consumption concerning humidity where if the humidity of the current day is low, then the average energy demand will be high. Because of targeting load forecasting in London, the trends of the electric data are very different from Asian countries. Their electricity needs increase in the winter, instead of summer. The unit of humidity is g/kg per day, and the average energy unit is Kwh.

This study tried to present the trends of energy consumption *vs* cloud cover. In most research articles (*Dong, Qian & Huang, 2017b*; *Dong, Yassine & Armitage, 2020*) there were just two main weather features for the load prediction: temperature and humidity. In Fig. 8 the trend, of cloud cover concerning energy is shown. In contrast, in this research, tried to add more features that have a close relationship with the average energy consumption of that day. In this trend, it is observed that if the cloud cover of that day goes low, then the average energy of that day goes high. The movements of cloud cover concerning power show the behavior of the data set from 2011 to 2014. The unit of the cloud cover is okta, and the Unit of average energy is Kwh.

In Fig. 9, the data-sets trends show the graph of average energy *vs* visibility. The visibility of the season or the weather can directly impact load forecasting. For example, when the weather is a bit cloudy, there must be low visibility of light there. There must be a

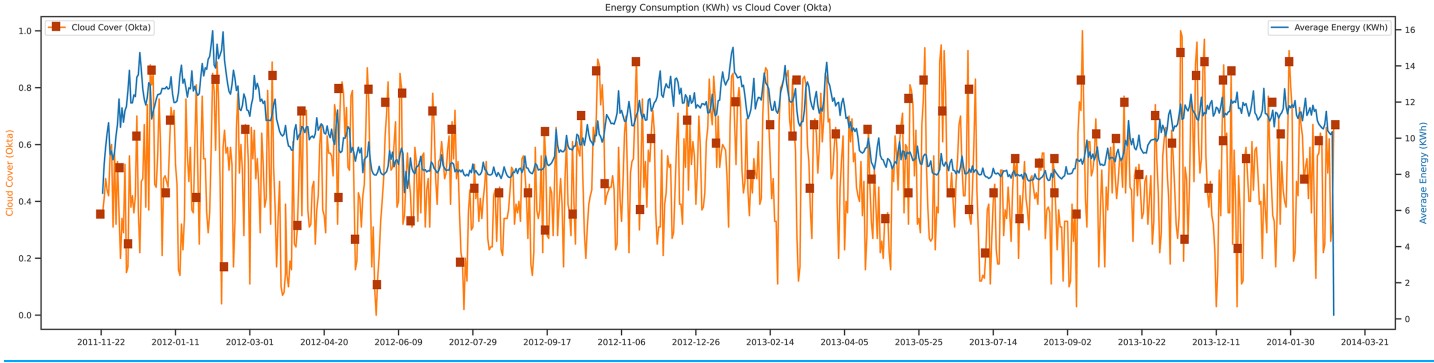

**Figure 8  Energy consumption *vs* cloud cover.**               

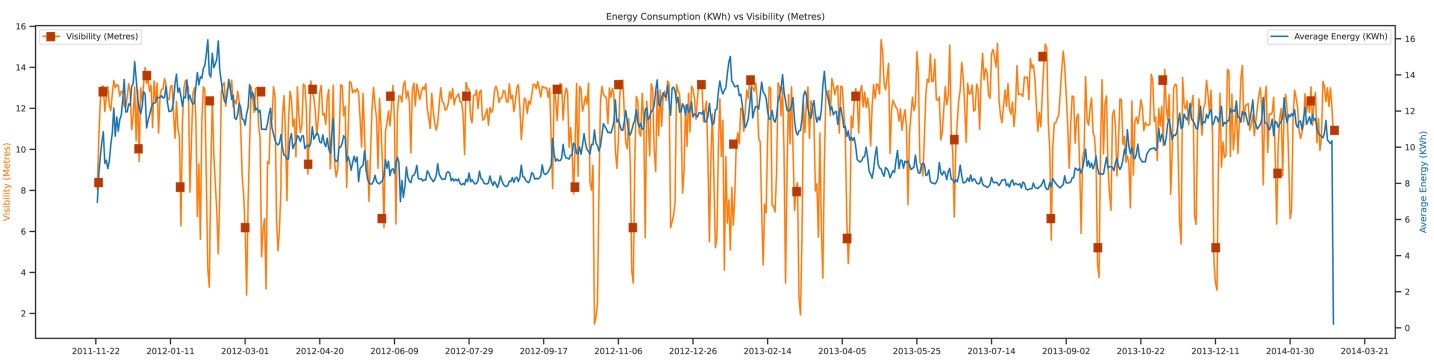

**Figure 9  Energy consumption *vs* visibility.**               

need for a higher electric city than the standard days. The trends of data show from September 2011 to March 2014. The unit of visibility is measured in terms of meters. The average energy unit can be calculated in KWh. The trends are shown in the figure; as the visibility goes higher, the average energy consumption of that day is low compared to other days.

Figure 10, shows wind speed *vs* average energy trend. The wind speed change occurs due to a change in the temperature, which is also an essential factor in future load prediction. The wind speed trend shown in the figure; shows that when the wind speed of a specific region is low, then the average energy of that day will be high compared to other days. The unit of wind speed is KM/h, and the energy unit is Kwh. It shows the complete trend of almost three years of the data set.

The UV index *vs* energy consumption trend is presented. The UV index is an essential feature in future load prediction. If you have to work outside (farms, fields, *etc.*), you first check the UV index of that day. If the UV index is high, it can damage the skin and also can harm the ecosystem. The UV index scale that we represent in our data set can be a scale from 0 to 11, whereas an average of above 6 UV index can harm human life and increase the average energy consumption of that day. This feature is used for load prediction in the proposed study. Figure 11, depicts the graphical representation from 2011 to 2014.

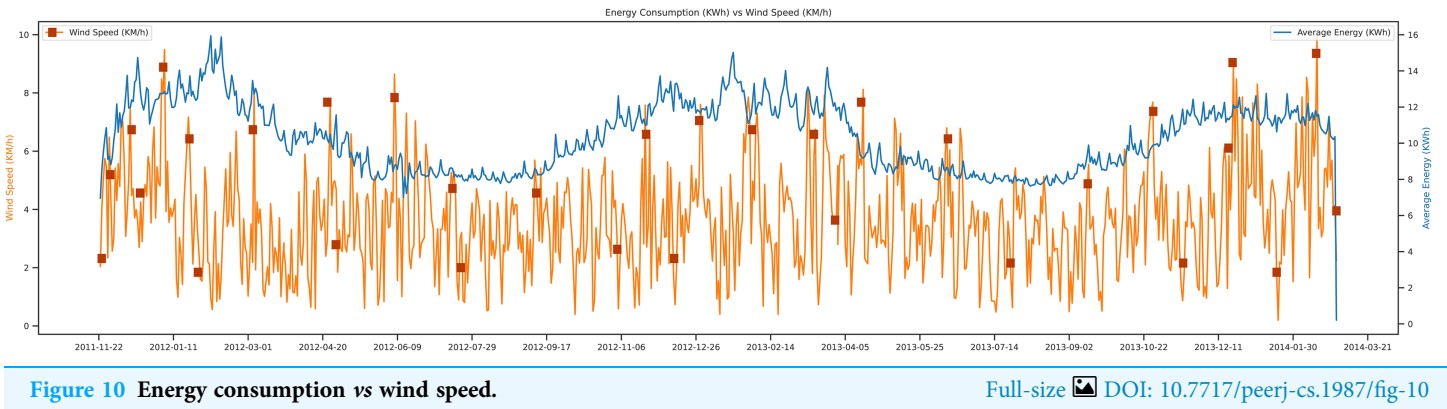

**Figure 10  Energy consumption *vs* wind speed.**

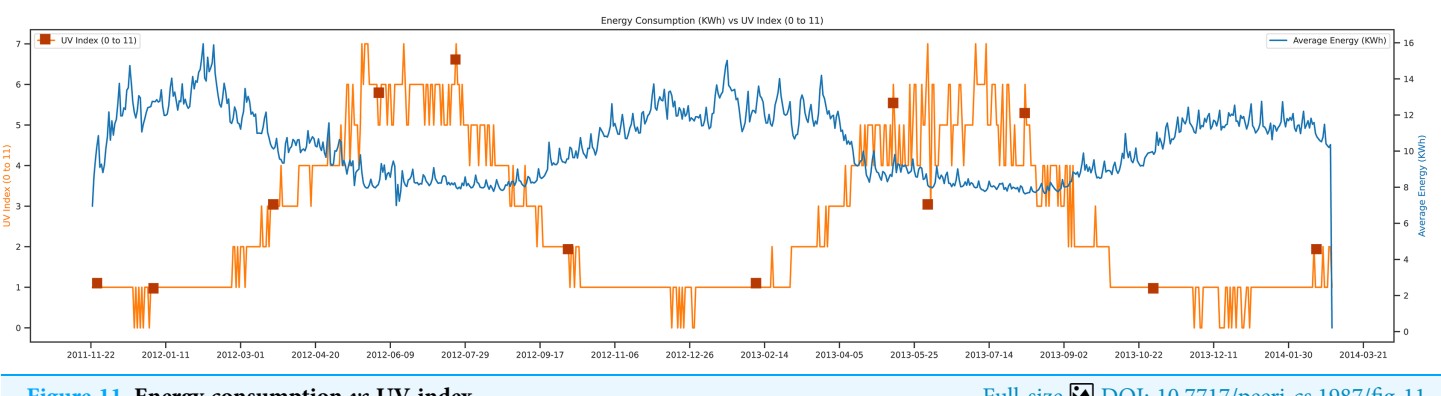

**Figure 11  Energy consumption *vs* UV index.**

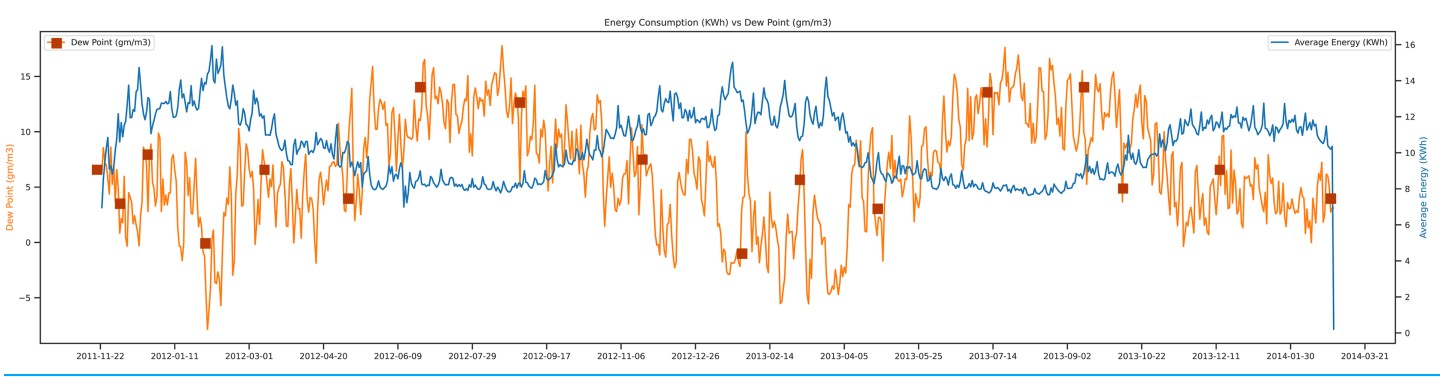

**Figure 12  Energy consumption *vs* dew points.**

The vital feature used in this study is the dew factor, which can play a vital role in the prediction. The dew point of a current day is associated with the temperature of that day. Most of the time, it can be important in the consumption of electricity in the winter. In Fig. 12, it can be observed that when the dew point is increasing on a particular day, the

consumption of electricity data will be minimal. The unit of dew point is gm/m, and the unit of electricity is Kwh.

## EXPERIMENTAL RESULTS AND DISCUSSION

To conduct short-term load forecasting in smart grids, this study implemented many machine learning-based approaches and handled smart meters data (*Daignan, 2014*), the weather data (*Daignan, 2014*), and holidays data. Furthermore, the proposed approach discusses the experiments that have been conducted so far. The data that is used is continuous therefore, the regression model is applied. The load values are ongoing and can not map into binary.

For research purposes, three main features from the data set were selected, which are LClid, day, and energy max to calculate the average energy of a day.

In the second step of implementation, the energy and weather data set is pre-processed. Later, remove the negative values, and redundant data from both the weather and the energy data set and combine the data according to the date. For research purposes, electric data with weather data is mapped.

The correlation graph in Fig. 13 is the graphical representation of many different correlations between different components. The method selected for the correlation is known as a heat map. From the figure, it can be seen that features are highly correlated with each other. Here 0.8 value of the correlation as a highly correlated feature is selected. In the literature review, several studies have selected two main weather features: temperature and humidity. Instead of these two features, this study encompasses eight more weather features that can directly affect the electric load in a smart grid system.

Another data set that used for load prediction is the 'holiday data'. In the holiday data set, there were 26 records. The holidays can impact the future load prediction as shown in Fig. 13 the trend of electric usage on weekdays and weekdays shows the different trends. On average days, the load usage trends of electricity are different from the holidays. In this data set (*Daignan, 2014*), the holiday data set has 26 records for a year. The holiday data that are using is from London. It contains information on holidays, such as the ones utilized for this research. It might indicate whether the next day is a holiday or just a common day. The value for a holiday is set to 1, and the value for a typical day is set to 0.

The experimental findings can be divided into two methods, each evaluating the performance of the baseline techniques with the proposed approach using the MAPE, RMSE, NMAE, MSLE, and R2 scores. In the results of all these, the proposed support vector regression model with some other baseline methods for comparison purposes.

Different machine learning methods were implemented like conventional neural networks, decision trees, random forests, K-nearest neighbor, and the linear regression model to verify the proposed model's effectiveness and accuracy with the state-of-the-art approaches as shown in graphs. As the data is continuous, therefore, implemented the most effective models for comparison. The actual *vs* predicted graph on the support vector regression model can be seen in Fig. 14 whereas, the linear regression model's performance can be seen in Fig. 15.

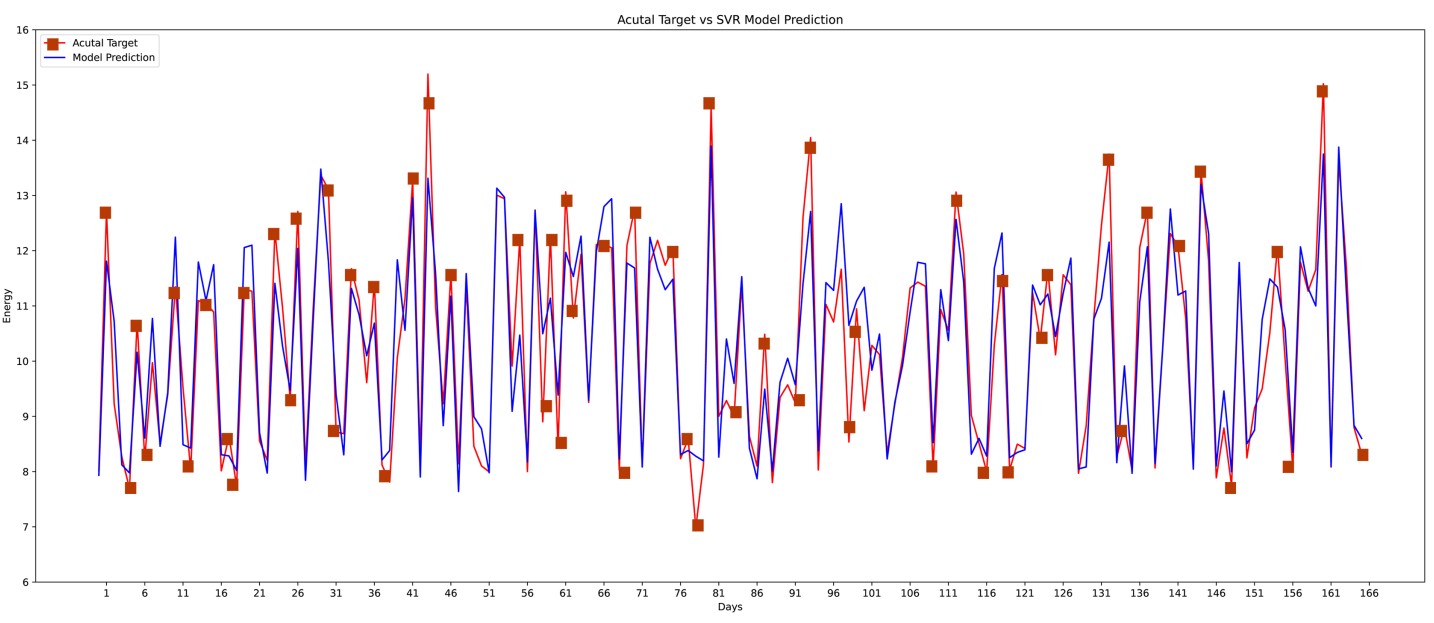

**Figure 13 Correlation graphs of features.**

**Figure 14 Support vector regression model actual *vs* predicted graph.**

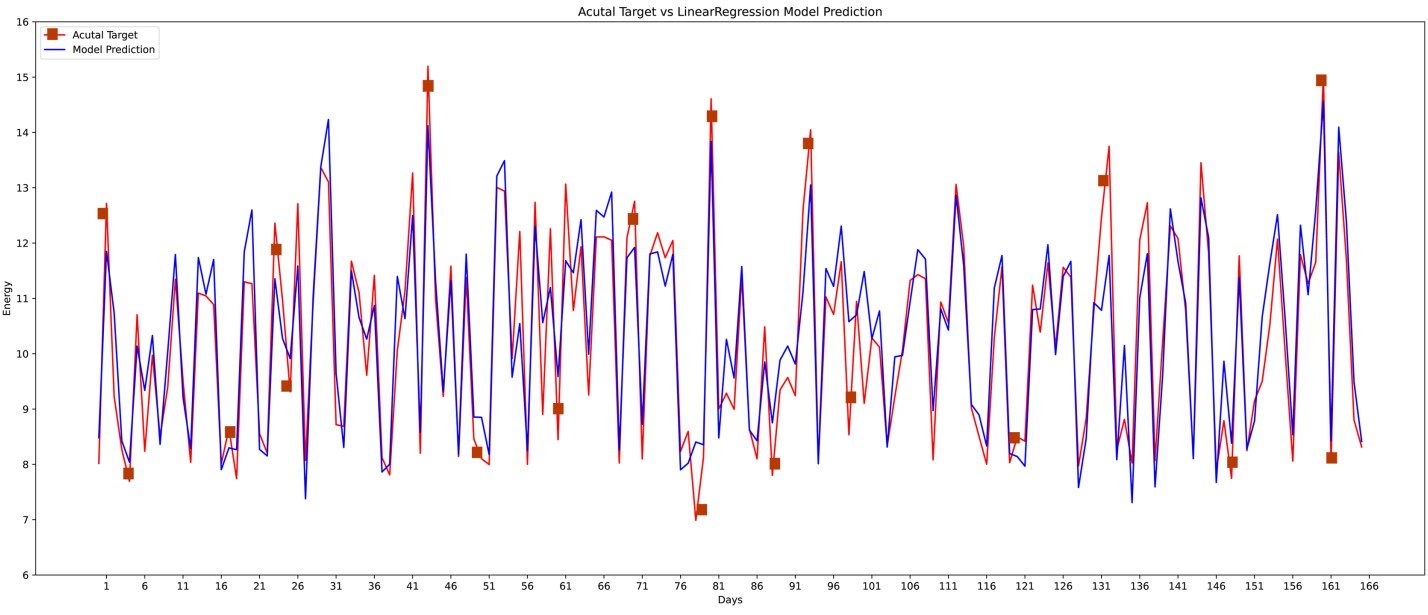

**Figure 15 Linear regression actual *vs* predicted graph.**

The third machine learning model that is implemented is the KNR regressor. KNR works calculating distance query instances data for linear regression problems. KNR is choosing K closest examples query and then voting most frequent labels case regression. The actual and the predicted value graph of the KNR model is shown in Fig. 16.

In the fourth model, the most prominent model is used, which is a conventional neural network as indicated in Fig. 17 that accepts the data in three dimensions. For applying the CNN model the data into three dimensions is applied to it as the CNN+k means were used for the short-term load predictions (*Dong, Qian & Huang, 2017b*). The CNN model performs well compared to other machine learning classifiers. The structure of CNN is made up of two layers. The first is the feature extraction layer, which connects each neuron's input to the preceding layer's local receptive fields and extracts the local feature.

The fifth model that is implemented on this data set is the random forest which is the most widely used machine learning model. The graphical representation of actual and predicted load through the random forest can be shown in Fig. 18. Random forest is a learning method that is supervised. It creates a "forest" out of an ensemble of decision trees, typically trained using the "bagging" technique. The bagging method's basic premise is that combining several learning models improves the outcome. The random forest algorithm performs well as compared to another classifier just because of the bagging method.

The sixth algorithm that is used in this research comparison is the decision tree (see Fig. 19). The decision tree shows its effectiveness somehow and performs well in some cases. Decision trees use various methods. The homogeneity of the resulting sub-nodes is

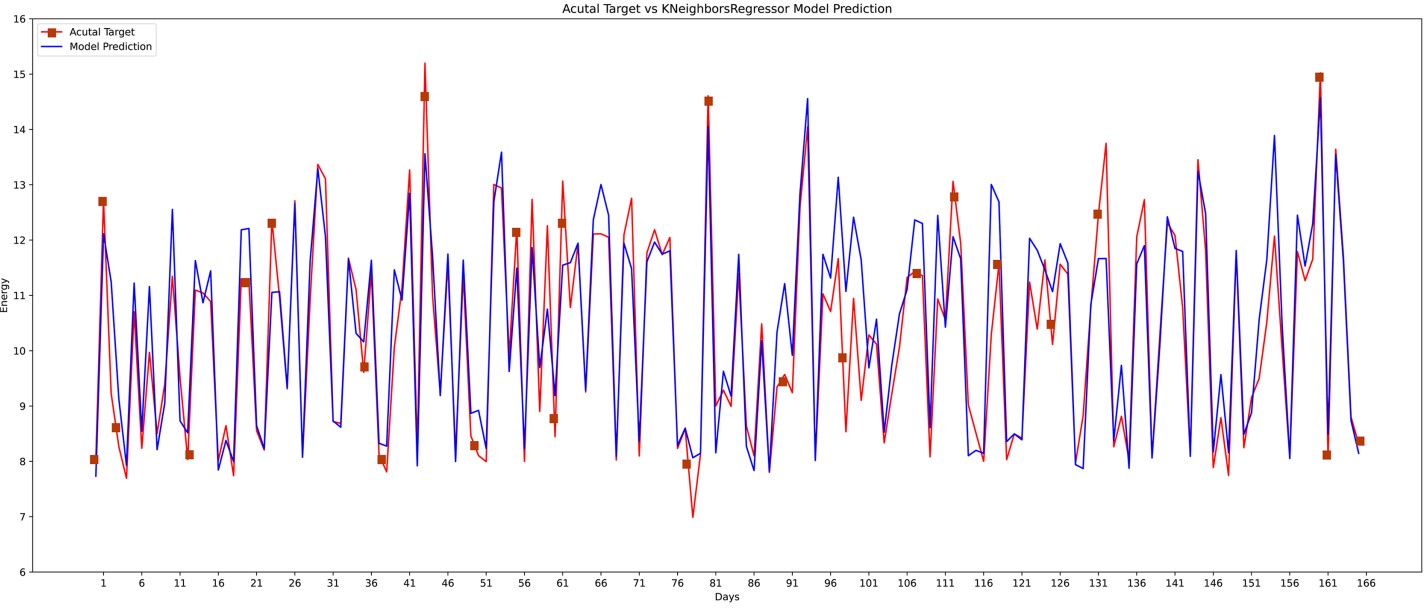

**Figure 16 K-nearest neighbor actual *vs* predicted graph.**     

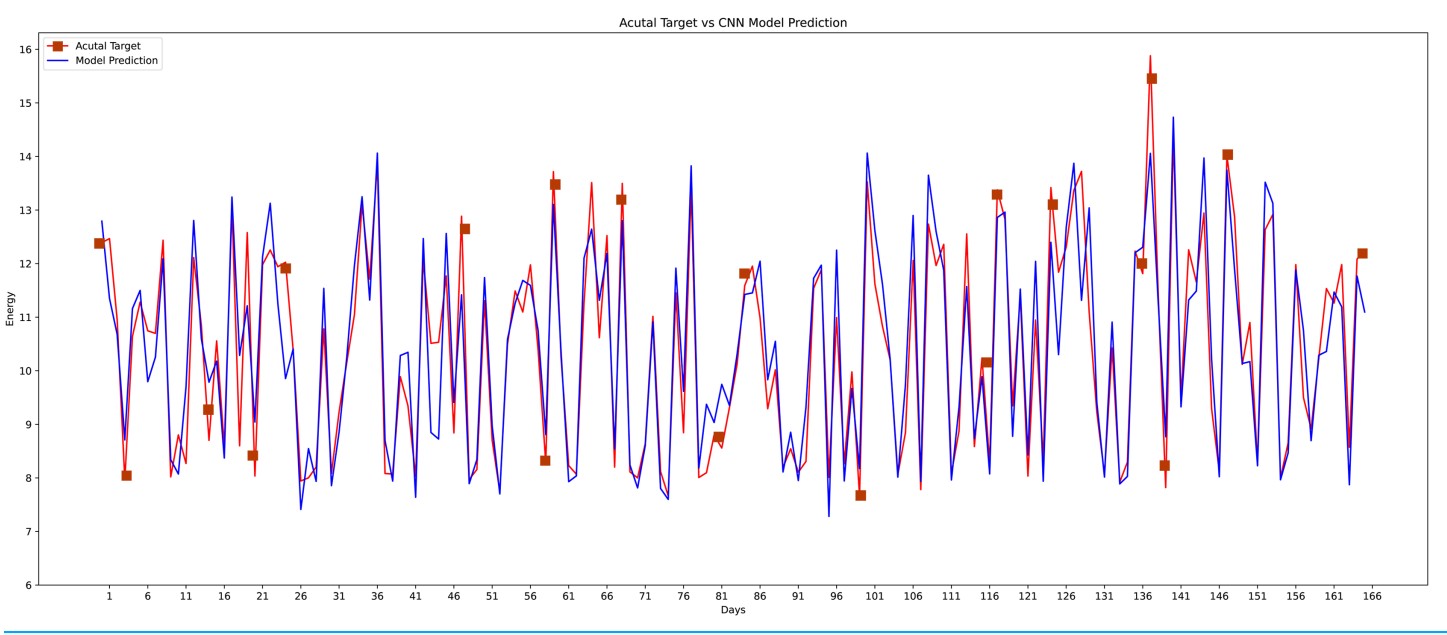

**Figure 17 Conventional neural network actual *vs* predicted graph.**     

increased when sub-nodes are created. The decision tree divides the nodes into sub-nodes based on all available factors and then chooses the split that produces the most homogeneous sub-nodes.

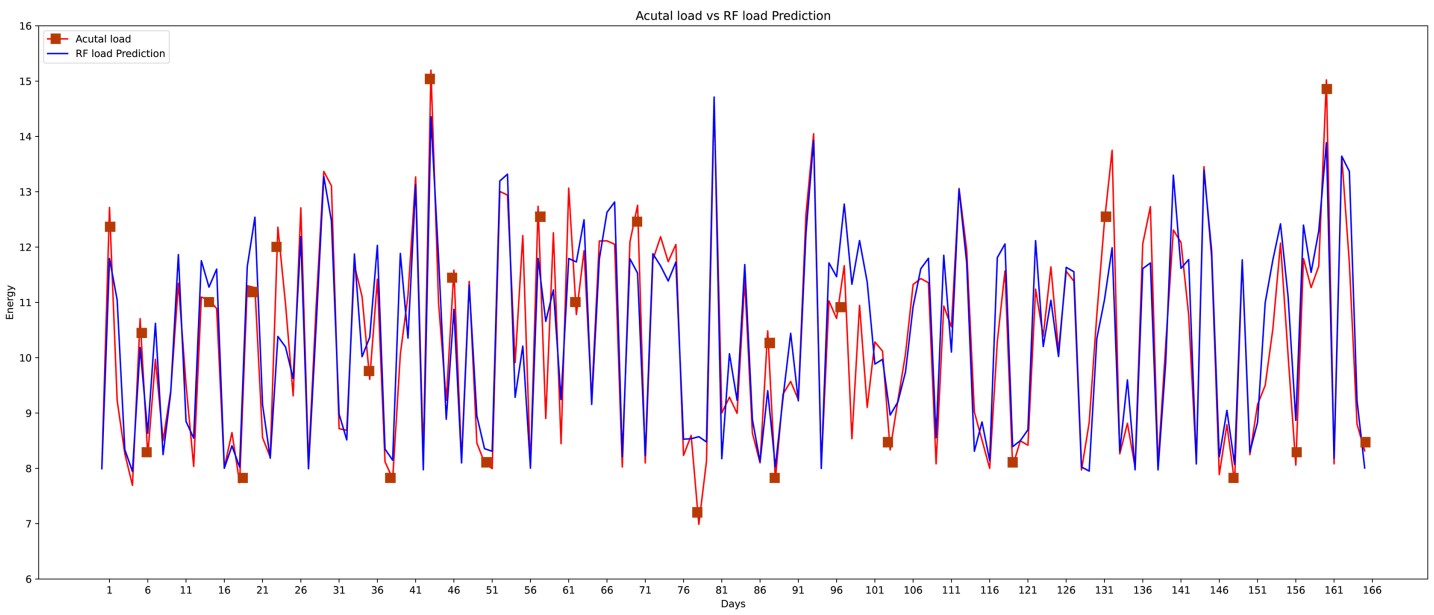

**Figure 18 Random forest actual *vs* predicted graph.**

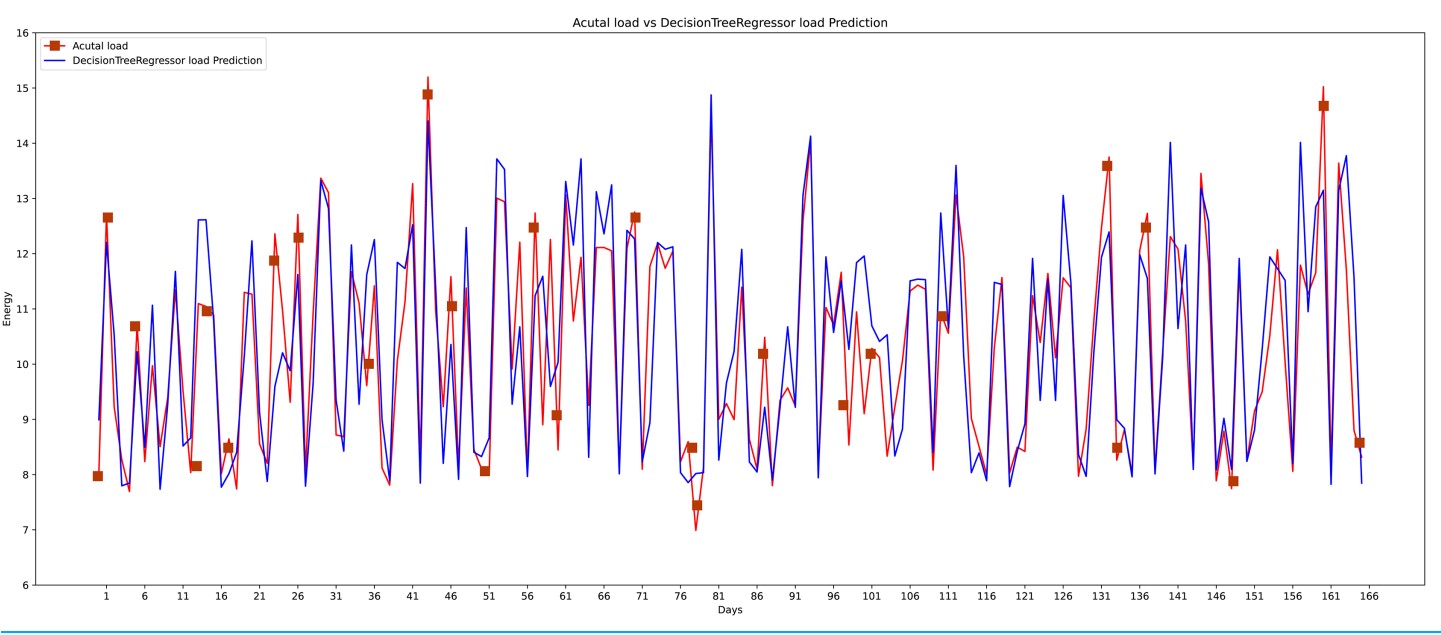

**Figure 19 Decision tree actual *vs* predicted graph.**

## Comparison with different machine learning techniques

The proposed method is constructed based on SVR. Therefore, the proposed model first compares this approach with the SVR model. The best MAPE values obtained by the proposed methods are 5.055 as shown in Fig. 20, which means that the proposed method outperforms other CNN-based load forecasting methods with the help of clustering

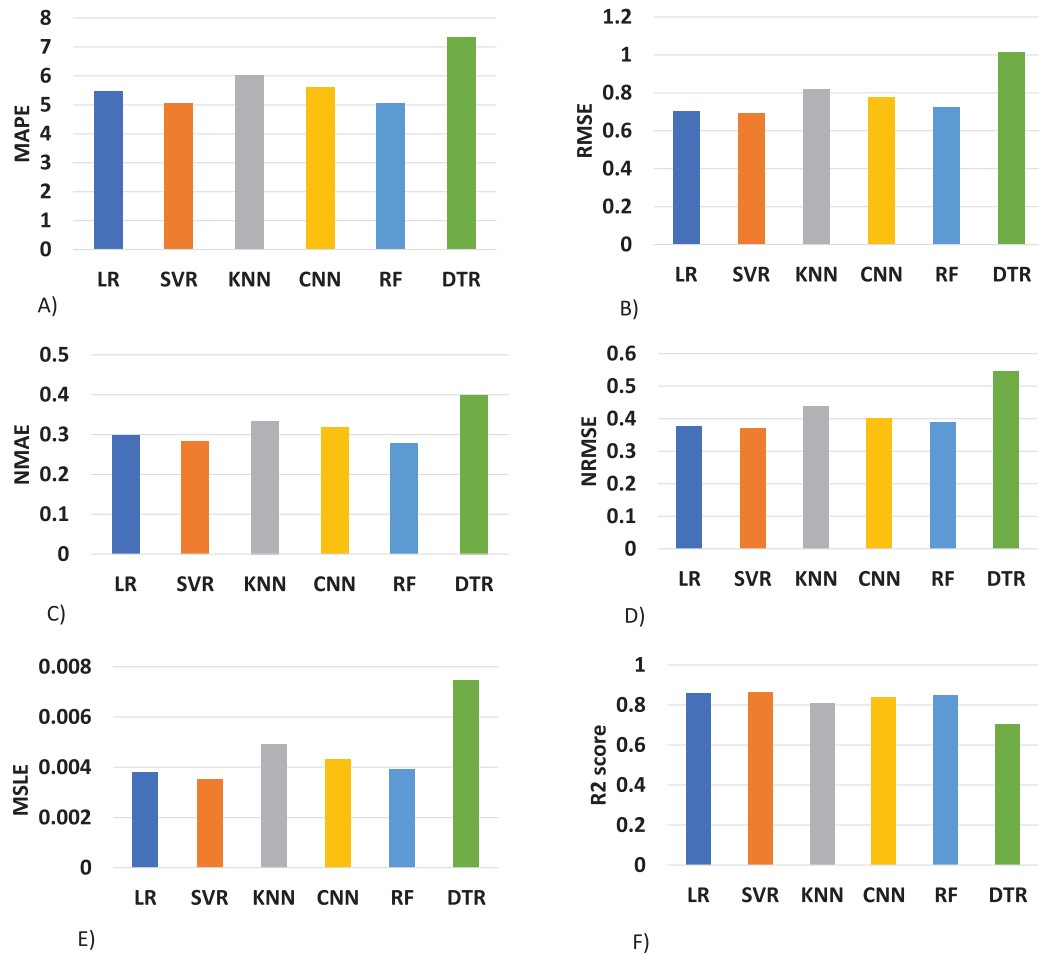

**Figure 20 Comparison of results of different (ML) evaluation matrices.** (A) MAPE evaluations with other machine learning methods. (B) RMSE evaluations with other machine learning methods. (C) NMAE evaluations with other machine learning methods. (D) NRMSE evaluations with other machine learning methods. (E) MLSE evaluations with other machine learning methods. (F) R2 score evaluations with other machine learning methods.

techniques. Considering MAPE Fig. 20A, RMSE Fig. 20B, NMAE Fig. 20C, NRMSE Fig. 20D, MSLE Fig. 20E and R2 scores Fig. 20F, the proposed approach can also enhance performance. In other words, clustering on data sets can improve performance distinctly. Also, the proposed study selected some more weather features that have direct effects on energy consumption. The support vector regression used in this research got better results than other machine learning-based techniques. Figure 14, directly examines the model, trying to cover the real load graphs but not approaching the actual load curve. It is a sign that the proposed model is performing well, and it did not show the over-fitting issues.

The value of the support vector regression is less than other machine learning techniques. In the mean absolute percentage error SVR model is performing well as compared to other techniques. The less value of mean absolute error means higher accuracy. The SVR model is performing well at 5% better than linear regression, 12% better than K-nearest neighbor, 7% better than a conventional neural network, 0.15% better than

**Table 4 Comparison of different machine learning models in tabular form.**

| Name of machine learning model | Mean absolute percentage error (MAPE) | Root mean square error (RMSE) | Normalized mean absolute error (NMAE) | Normalized root mean square error (NRMSE) | Mean squared logarithmic error (MSLE) | Coefficient of determination (R2 score) |
|---|---|---|---|---|---|---|
| Support vector regression | 5.055 | 0.692 | 0.281 | 0.371 | 0.003 | 0.003 |
| Linear regression | 5.480 | 0.703 | 0.297 | 0.377 | 0.003 | 0.856 |
| K-nearest neighbor | 6.036 | 0.818 | 0.334 | 0.438 | 0.004 | 0.806 |
| Convolutional neural network | 5.169 | 0.779 | 0.316 | 0.400 | 0.004 | 0.838 |
| Random forest | 5.061 | 0.723 | 0.278 | 0.388 | 0.003 | 0.848 |
| Decision tree regressor | 7.351 | 1.011 | 0.399 | 0.544 | 0.007 | 0.701 |

random forest, and 28% better than the decision tree model. The random forest is also performing well in this scenario as compared to other techniques.

In Fig. 20B, we compared the root mean square error of the different models implemented on our data set. The SVR model performs 1% better then than linear regression, 10% better than the K-nearest neighbor, 7% better than the conventional neural network, 3% better than random forest, and 27% better than a decision tree. in this also, the lower value of RMSE refers to the higher accuracy.

In Fig. 20C, the result of NMAE is presented. SVR less performs than RF in term of normalized mean absolute error compared to other baseline techniques. Our SVR model is performing well as compared to other models 2% better than LR, 11% better than the KNN, 6% better than the CNN, and 24% better then than DTR. The SVR model shows the degraded performance in this as compared to a random forest. The random forest performs 2% better than SVR.

In Fig. 20D, normalized root mean square error is compared with different techniques and found that the SVR performs 16% better than KNN, 16% CNN, 13% better than RF, 40% better than DTR. In NRMSE the linear regression produces a comparable result to the SVR model.

In Fig. 20E, a new evaluation matrix for this research work is used. Mean squared logarithmic error (MSLE) shows the difference between actual and predicted values. In terms of MSLE, SVR is performing 4% better than LR, 18% better than KNN, 10% better than CNN, 5% better than RF, and 49% better than the DTR.

In Fig. 20F, results are compared with the essential R2 score. We also found that the proposed support vector regression is performing 1% better than LR, 6% better than KNN, 3% better than CNN, 2% better than RF, AND 16% better than the DTR.

The finding of this article concludes that pre-processing the data and removing the outlier from the data can gives a better prediction of load without getting the over-fitted results. Furthermore, the proposed three-tier load forecasting scheme in the smart grids

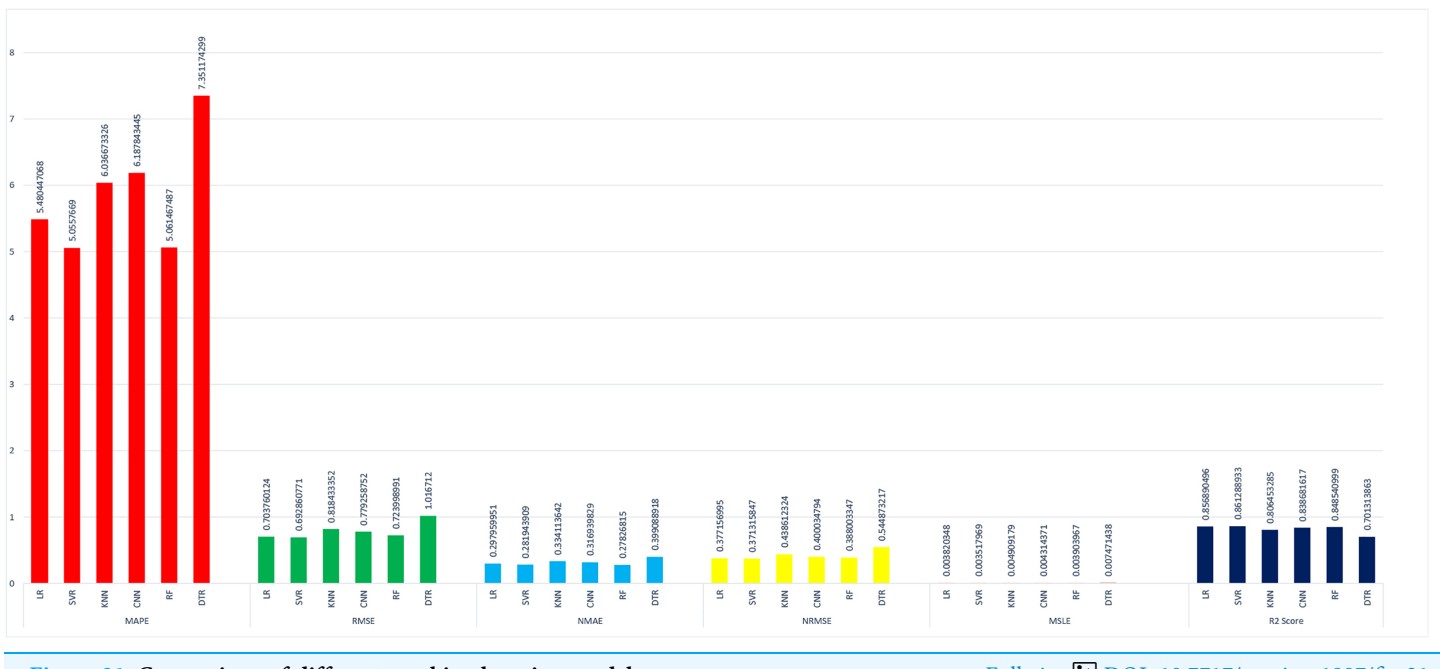

**Figure 21  Comparison of different machine learning models.**

including the data of weather features in this study is a vital role that had not been used previously in the literature. Lastly, results were compared with other machine learning models that came up with better accuracy as compared to the other machine learning approaches.

The details results of SVR concerning other machine learning approaches are shown in tabular form in Table 4 and in Fig. 21. Where the data is visualized separately for comparison purposes. The evaluation metrics are used for comparing the algorithms *i.e.*, MAPE (*Dong, Qian & Huang, 2017b*), RMSE (*Dong, Qian & Huang, 2017b*), NMAE (*Dong, Qian & Huang, 2017b*), NRMSE (*Dong, Qian & Huang, 2017b*), MSLE (*Khan et al., 2019*) and R2 (*Khan et al., 2019*). In this Fig. 21 the lesser value of the bar graph means high accuracy. In our last R2 score metrics the higher prediction accuracy has a higher bar value.

## CONCLUSIONS

In the proposed approach, we have used several weather features that have not been used previously to predict the load and trends in energy consumption. Furthermore, this study solved the problem of the overfitting issue by selecting the best-fitted subsets of the features that are not correlated with the energy and other features. Also, included the holiday data in this research to get a better prediction of load on the holidays as well. For this several experiments were conducted and found that the support vector regression performed well. Results of R2 score values such as 5.055 Mape and the 0.69 RMSE, 0.37 NRMSE, 0.0072 MSLE, and 0.86 were attained. It is better than other baseline methods. This study uses a real-world power industry data set with over 1.4 million load records. The experimental findings show that the method is the effect of the proposed method. In the future, several

other features that are directly associated with the energy in an environment can be considered. Moreover, additional features like user behavior, the country's political condition, and other related factors can be utilized.

### Funding
The authors received no funding for this work.

### Competing Interests
Muhammad Aleem is an Academic Editor for PeerJ.

### Author Contributions
- Muhammad Yasir Masood conceived and designed the experiments, performed the experiments, performed the computation work, prepared figures and/or tables, authored or reviewed drafts of the article, and approved the final draft.
- Sana Aurangzeb conceived and designed the experiments, performed the experiments, performed the computation work, prepared figures and/or tables, authored or reviewed drafts of the article, and approved the final draft.
- Muhammad Aleem conceived and designed the experiments, performed the experiments, analyzed the data, authored or reviewed drafts of the article, and approved the final draft.
- Ameen Chilwan performed the experiments, analyzed the data, authored or reviewed drafts of the article, and approved the final draft.
- Muhammad Awais analyzed the data, performed the computation work, prepared figures and/or tables, and approved the final draft.

### Data Availability
The Smart meters in London data is available at Kaggle: https://www.kaggle.com/datasets/jeanmidev/smart-meters-in-london?select=darksky_parameters_documentation.html.

### Supplemental Information
Supplemental information for this article can be found online at http://dx.doi.org/10.7717/peerj-cs.1987#supplemental-information.

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
