# Peer review of "Demand-side load forecasting in smart grids using machine learning techniques"

_PeerJ Computer Science, doi:10.7717/peerj-cs.1987_

## Round 0.1 · original submission · Major Revisions

I recommend that authors consider the following issues carefully. These suggestions aim to enhance the overall quality and impact of your research article.

Improve grammar and clarity for better understanding. Strengthen the literature study to demonstrate expertise. Enhance the quality of figures for clearer visualization. Explain the research problem in more detail. Be concise and avoid unnecessary explanations and repetition. Clarify the relevance of existing methods in your study. Present experimental results in tables for efficiency. Include crucial data like training and test errors.

**Language Note:** The Academic Editor has identified that the English language must be improved. PeerJ can provide language editing services - please contact us at copyediting@peerj.com for pricing (be sure to provide your manuscript number and title). Alternatively, you should make your own arrangements to improve the language quality and provide details in your response letter. – PeerJ Staff

Reviewer 1 ·

Basic reporting

In general, the language of the article seems understandable.
There are some grammatical errors. This reduces intelligibility.
A literature study is sufficient.
The quality of the figures should be increased.
The problem studied should be explained better.
There are too many unnecessary explanations in the article. There are also many repetitive sentences. Therefore, the intelligibility of the subject decreases.

Experimental design

The article does not contain much innovation for this journal. It is more suitable for magazines on energy and electricity.
Evaluated classification and estimation methods are already available in the literature. It is therefore not included for the purpose of this journal.
Experimental studies and explanations of results are not efficient. It would be better to discuss the results by presenting the results in a tabular form.
The values of the methods such as training and test errors are not given.

Validity of the findings

The original side of the article should be given better.

Additional comments

The article is unnecessarily too long. The original side should be emphasized directly by giving the missing aspects in the literature.
The methods used in estimation are already used in the literature. Therefore, it will not contribute to this journal.
It may be better to evaluate it in energy-related journals.

Cite this review as

·

Basic reporting

In the study entitled “Demand-side load forecasting in smart grids using machine learning techniques”, the authors address overfitting issues using a three-tier architecture that includes cloud layer, fog layer, and edge servers. In the approach proposed by the authors, more weather features that were not previously used were used primarily to estimate the load.

Experimental design

As a reviewer, I examined the work in detail. It is a well prepared and fluent work. In the study, a detailed literature review was made, the scientific deficiency was revealed, the method was developed to eliminate the deficiency and the results were analyzed in detail. I think that the study will make a positive contribution to the literature. It would be better if the starting point of the study, its motivation and its contribution to the literature are emphasized in the abstract part. At the end of the Related works section, the original aspect of this study and the main problem of the study should be emphasized.

Validity of the findings

The findings and comparison tables obtained as a result of the study are accurate and correct.

Reviewer 3 ·

Basic reporting

It appears you've described a research approach for electrical load forecasting, with a focus on using IoT devices, smart meters, and a three-tier architecture to improve the accuracy of load predictions.

Daily Consumption Electrical Networks (DCEN): This network appears to preprocess and extract relevant features from the collected data to provide input for load forecasting.

Intra Load Forecasting Networks (ILFN): This network performs the actual load forecasting based on the processed input data.

Your system is designed with a three-tier architecture, which includes cloud, fog, and edge servers. This architecture likely helps distribute the computational load and data processing tasks efficiently, especially considering the large volume of data involved in load forecasting.

Your approach utilizes more weather features than previously used in load prediction.

You provided several performance metrics to evaluate the effectiveness of your proposed method, including Mean Absolute Percentage Error (MAPE), Root Mean Square Error (RMSE), Normalized Mean Square Error (NRMSE), Mean Squared Logarithmic Error (MSLE), and R2 score values. These metrics are commonly used to assess the accuracy and quality of load forecasting models.

In conclusion, your proposed approach appears to be a comprehensive and data-driven solution to address the challenges of electrical load forecasting. The use of IoT devices, smart meters, a three-tier architecture, and SVR as the forecasting method demonstrates a commitment to improving accuracy and efficiency in load prediction.

Experimental design

Overall, it seems like you're taking a data-driven and comprehensive approach to short-term load forecasting in smart grids. By incorporating various data sources and considering user-specific factors, you're likely to improve the accuracy of your load forecasts and adapt to the changing dynamics of the electric grid over time. Machine learning-based regression models are a suitable choice for handling continuous load data and making predictions based on the available information.

Your approach appears to be thorough and well-documented, covering data preprocessing, feature selection, model selection, and model evaluation. The use of multiple machine learning models and the comparison of their performance against baseline methods provide valuable insights into the effectiveness of your proposed load forecasting model. Additionally, the incorporation of holiday data and a wide range of weather features should contribute to more accurate load predictions.

Your research appears to demonstrate the effectiveness of your proposed approach and the superiority of the SVR model in this context, supported by comprehensive evaluation metrics and graphical representations.

Validity of the findings

In conclusion, your research presents a comprehensive approach to load forecasting in smart grids, leveraging new weather features, addressing overfitting, and incorporating holiday data. The positive results achieved with SVR and the use of real-world data demonstrate the effectiveness of your proposed method, offering valuable insights for improving load forecasting in practical energy systems. Your plans for future work also show a commitment to continuous improvement and innovation in the field.

Additional comments

When I checked your references, I saw that articles from 2020 and previous years were reviewed. Comparisons with current articles will make this study more valuable.

Cite this review as

---

## Round 0.2 · Major Revisions

According to the referee report, it seems that there are still many areas that need improvement in your article. Addressing these points one by one will improve the quality of your article and provide you with an opportunity for re-evaluation.

Reviewer 1 ·

Basic reporting

- The wording of the article has been corrected. However, there is still a repetitive narrative. Previously given information is constantly repeated. There should be an arrangement at this stage.
- Personal identifiers are used a lot. For example, "we, our". This type of expression is not often encountered in an academic article. It should be more of a passive expression. For example, instead of saying "We made some innovations in this study.", instead of saying "Some innovations were made in this study." should be used. There should be present tense narration rather than past tense narration. Academic expression should be used instead of daily spoken language.
- Literature research should be improved. In this study, more recent studies should be examined, and articles in journals included in reputable indexes should be taken as reference rather than conference proceedings.
- Formal results should include clear definitions and detailed proofs of all terms and theorems.

Experimental design

In the study, it is stated that, unlike the literature, weather data is increased. So the number of features has been increased. Many estimation methods have been used and the results are discussed.

Validity of the findings

The data used in the study were taken from certain sources and are stated in the references. However, the data used is data obtained between 2011 and 2014. How will the use of data from approximately 10 years ago affect the reliability of the study?

The results are given with graphs and tables. Here, the dimensions of the matrices used in the training and estimation stages of the estimation methods can also be given.

Additional comments

It is mentioned in the study that, unlike the literature, different weather conditions and holidays are included in the forecast data. Weather will be more effective in networks with renewable energy sources. What is the proportion of renewable energy sources in the system discussed here? Are there enough renewable energy sources in the system to affect the results?

A separate method section should be opened in the study, and the method used and the characteristics of the data should be explained in detail here. Otherwise, the importance of the method used is not understood.

Most of the methods used are old methods, and the method that gives the best results in the study is the support vector regression method, which is a well-known method. Although the journal is a computer science journal, how can originality be achieved with these old methods? There should also be an innovation in the method section.

As can be seen, a prediction was made using only winter data. However, there will be an increase in load due to cooling systems during the summer months. Why were summer data not used?

The narrative of the article has improved, but there is still unnecessary and excessive explanation. Unnecessary repetitions should be avoided. Everyday language should not be used. The narrative should be sequential and orderly. The narrative must have a certain flow.

As it is understood, input data to ILFN is produced with DCEN. This situation would be better explained with a diagram.

Table 1 and Table 2 are not suitable for use.

Cite this review as

---

## Round 0.3 · Minor Revisions

In the referee's recommendation, "The rate of articles in the literature research should be further increased." It is said.
I recommend revision in this direction.

Reviewer 1 ·

Basic reporting

The rate of articles in literature research should be further increased.

Experimental design

Most of the necessary arrangements have been made.

Validity of the findings

Most of the necessary arrangements have been made.

Additional comments

The proportion of renewable energy sources in the grid should be given.

Cite this review as

---

## Round 0.4 · accepted · Accept

After thorough academic evaluations, it was concluded that the article met the required criteria. We sincerely appreciate your valuable contribution to the review process.